# ATTENTION PRIVILEGED REINFORCEMENT LEARNING FOR DOMAIN TRANSFER

## ABSTRACT

Applying reinforcement learning (RL) to physical systems presents notable challenges, given requirements regarding sample efficiency, safety, and physical constraints compared to simulated environments. To enable transfer of policies trained in simulation, randomising simulation parameters leads to more robust policies, but also significantly extends training time. In this paper, we exploit access to privileged information (such as environment states) often available in simulation, in order to improve and accelerate learning over randomised environments. We introduce Attention Privileged Reinforcement Learning (APRiL), which equips the agent with an attention mechanism and makes use of state information in simulation, learning to align attention between state- and image-based policies while additionally sharing generated data. During deployment we can apply the image-based policy to remove the requirement of access to additional information. We experimentally demonstrate accelerated and more robust learning on a number of diverse domains, leading to improved final performance for environments both within and outside the training distribution. [1].

## 1 INTRODUCTION

Deep Reinforcement Learning (RL) has recently provided significant successes in a range of areas, including video games (Mnih et al., 2015), board games (Silver et al., 2017), simulated continuous control tasks (Lillicrap et al., 2015), and robotic manipulation (Haarnoja et al., 2018; Haarnoja, 2018; Riedmiller et al., 2018; OpenAI et al., 2018; Schwab et al., 2019; Andrychowicz et al., 2017). However, application to physical systems has proven to be challenging in general, due to expensive and slow data generation as well as safety challenges when running untrained policies. A common approach to circumvent these issues is to transfer models trained in simulation to the real world (Tobin et al., 2017; Rusu et al., 2016; Held et al., 2017). However, simulators only represent approximations of a physical system. Due to physical, visual, and behavioural discrepancies, naively transferring RL agents trained in simulation onto the real world can be challenging.

To bridge the gap between simulation and the real world, we can either aim to align both domains (Ganin et al., 2016; Bousmalis et al., 2016; Wulfmeier et al., 2017) or ensure that the real system is covered by the distribution of simulated training data (OpenAI et al., 2018; Tobin et al., 2017; Pinto et al., 2018; Sadeghi & Levine, 2016; Viereck et al., 2017). However, training under a distribution of randomised visual attributes of the simulator, such as textures and lighting (Sadeghi & Levine, 2016; Viereck et al., 2017), as well as physics (OpenAI et al., 2018), can be substantially more difficult and slower due to the increased variability of the learning domain (OpenAI et al., 2018; Tobin et al., 2017).

The more structured and informative the input representation is with respect to the task, the quicker the agent can be trained. A clear example of this effect can be found when an agent is trained with image inputs, versus training with access to the exact simulator states (Tassa et al., 2018; Pinto et al., 2018). However, visual perception is more general and access to more compressed representations can often be limited. When exact states are available during training but not deployment, we can make use of information asymmetric actor-critic methods (Pinto et al., 2018; Schwab et al., 2019) to train the critic faster via access to the state while providing only images for the actor.

---

[1] Videos comparing the policy behaviours of APRiL to the asymmetric DDPG baseline can be found here

By introducing Attention Privileged Reinforcement Learning (APRiL), we aim to further leverage access to exact states. APRiL leverages states not only to train the critic, but indirectly also for an image-based actor. Extending asymmetric actor-critic methods, APRiL concurrently trains two actor-critic systems (one symmetric, state-based agent, and another asymmetric agent with image-dependent actor). Both actors utilise an attention mechanism to filter input data and by having access to the simulation rendering system, we can optimise image and state based attention masks to align.

By additionally sharing the replay buffer between both agents, we can accelerate the learning process of the image-based actor by training on better performing states that are more quickly discovered by the state-based actor due to its lower dimensional input that is invariant to visual randomisation.

The key benefits of APRiL lie in its application to domain transfer. When training with domain randomisation for transfer, bootstrapping via asymmetric information has displayed crucial benefits (Pinto et al., 2018). Visual randomisation substantially increases the complexity of the image-based actor's task. Under this setting, the attention network can support invariance with respect to the irrelevant, but highly varying, parts of the image. Furthermore, the convergence of the state-space actor remains unaffected by visual randomisation.

We experimentally demonstrate considerable improvements regarding learning convergence and more robust transfer on a set of continuous action domains including: 2D navigation, 2D locomotion and 3D robotic manipulation.

## 2 PROBLEM SETUP

Before introducing Attention Privileged Reinforcement Learning (APRiL), this section provides a background for the RL algorithms used. For a more in-depth introduction please refer to Lillicrap et al. (2015) and Pinto et al. (2018).

### 2.1 REINFORCEMENT LEARNING

We describe an agent's environment as a Partially Observable Markov Decision Process which is represented as the tuple $(S, O, A, P, r, \gamma, s_0)$, where $S$ denotes a set of continuous states, $A$ denotes a set of either discrete or continuous actions, $P : S \times A \times S \to \{x \in \mathbb{R} | 0 \leq x \leq 1\}$ is the transition probability function, $r : S \times A \to \mathbb{R}$ is the reward function, $\gamma$ is the discount factor, and $s_0$ is the initial state distribution. $O$ is a set of continuous observations corresponding to continuous states in $S$. At every time-step $t$, the agent takes action $a_t = \pi(\cdot | s_t)$ according to its policy $\pi : S \to A$. The policy is optimised as to maximize the expected return $R_t = E_{s_0}[\sum_{i=t}^{\infty} \gamma^{i-t} r_i | s_0]$. The agent's Q-function is defined as $Q_\pi(s_t, a_t) = E[R_t | s_t, a_t]$.

### 2.2 ASYMMETRIC DEEP DETERMINISTIC POLICY GRADIENTS

Asymmetric Deep Deterministic Policy Gradients (asymmetric DDPG) (Pinto et al., 2018) represents a type of actor-critic algorithm designed specifically for efficient learning of a deterministic, observation-based policy in simulation for sim-to-real transfer. This is achieved by leveraging access to more compressed, informative environment states, available in simulation, to speed up and stabilise training of the critic.

The algorithm maintains two neural networks: an observation-based actor or policy $\pi_\theta : O \to A$ (with parameters $\theta$) used during training and test time, and a state-based Q-function (also known as critic) $Q_\phi^\pi : S \times A \to R$ (with parameters $\phi$) which is only used during training.

To enable exploration, the method (like its symmetric version (Silver et al., 2014)) relies on a noisy version of the policy (called behavioural policy), e.g. $\pi_b(o) = \pi(o) + z$ where $z \sim \mathcal{N}(0, 1)$ (see Appendix C for our particular instantiation). The transition tuples $(s_t, o_t, a_t, r_t, s_{t+1}, o_{t+1})$ encountered during training are stored in a replay buffer (Mnih et al., 2015). Training examples sampled from the replay buffer are used to optimize the critic and actor. By minimizing the Bellman error loss $\mathcal{L}_{critic} = (Q(s_t, a_t) - y_t)^2$, where $y_t = r_t + \gamma Q(s_{t+1}, \pi(o_{t+1}))$, the critic is optimized to approximate the true Q values. The actor is optimized by minimizing the loss $\mathcal{L}_{actor} = -E_{s, o \sim \pi_b(o)}[Q(s, \pi(o))]$.

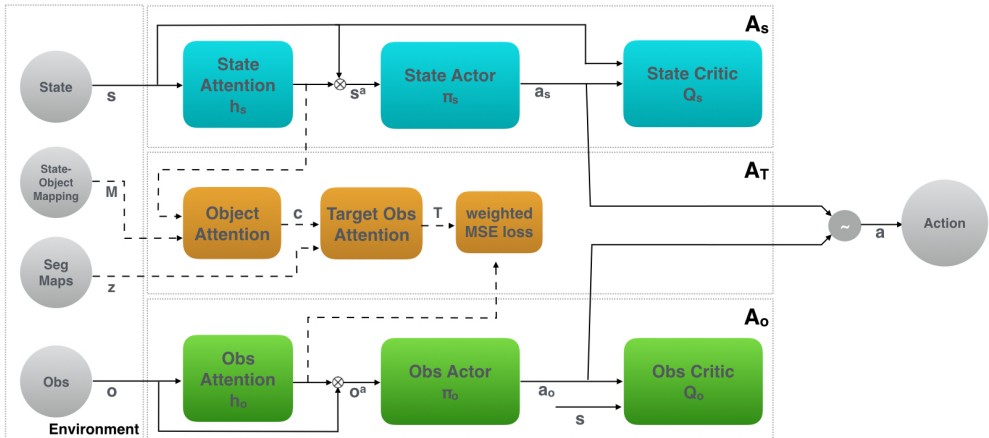

Figure 1: Attention Privileged Reinforcement Learning model structure. Dashed lines indicate attention alignment process. The $\sim$ operator signifies that experiences are evenly sampled from both agents. The $\otimes$ operator represents element-wise multiplication.

# 3 ATTENTION PRIVILEGED REINFORCEMENT LEARNING (APRiL)

APRiL proposes to improve the performance and sample efficiency of an observation-based agent by using a quicker learning actor that has access to exact environment states, sharing replay buffers, and aligning attention mechanisms between both actors. While we focus in the following sections on extending asymmetric DDPG (Pinto et al., 2018), these ideas are generally applicable to off-policy actor-critic methods (Konda & Tsitsiklis, 2000).

APRiL is comprised of three modules as displayed in Figure 1. The first two modules, $A_s$ and $A_o$, are actor-critic algorithms with an attention network incorporated over the input to each actor. For the *state-based* module $A_s$ we use standard symmetric DDPG, while the *observation-based* module $A_o$ builds on asymmetric DDPG. Finally, the third part $A_T$ represents the alignment process between attention mechanisms of both actor-critic agents to more effectively transfer knowledge between the quicker and slower learners, $A_s$ and $A_o$, respectively.

$A_s$ consists of three networks: $Q_s^\pi$, $\pi_s$, $h_s$ (respectively critic, actor, and attention) with parameters $\{\phi_s, \theta_s, \psi_s\}$. Given input state $s_t$, the attention network outputs a soft gating mask $h_t$ of same dimensionality as the input, with values ranging between $[0, 1]$. The input to the actor is an attention-filtered version of the state, $s_t^a = h_s(s_t) \odot s_t$. To encourage a sparse masking function, we found that training this attention module on both the traditional DDPG loss as well as an entropy loss helped:

$$\mathcal{L}_{h_s} = -E_{s \sim \pi_b}[Q_s(s, \pi_s(s^a)) - \beta H(h_s(s))], \quad (1)$$

where $\beta$ is a hyperparameter to weight the additional entropy objective, and $\pi_b$ is the behaviour policy used to obtain experience (in this case from a shared replay buffer). The actor and critic networks $\pi_s$ and $Q_s$ are trained with the symmetric DDPG actor and Bellman error losses respectively.

Within $A_T$, the state-attention obtained in $A_s$ is converted to corresponding observation-attention $T$ to act as a self-supervised target for the observation-based agent in $A_o$. This is achieved in a two-step process. First, state-attention $h_s(s)$ is converted into object-attention $c$, which specifies how task-relevant each object in the scene is. Second, object-attention is converted to observation-space attention by performing a weighted sum over object-specific segmentation maps:

$$c = M \cdot h_s(s) \quad (2) \qquad\qquad T = \sum_{i=0}^{N-1} c_i \cdot z_i \quad (3)$$

Here, $M \in \{0, 1\}^{N \times n_s}$ (where $n_s$ is the dimensionality of $s$) is an environment-specific, predefined adjacency matrix that maps the dimensions of $s$ to each corresponding object, and $c \in [0, 1]^N$ is

then an attention vector over the $N$ objects in the environment. $c_i$ corresponds to the $i^{th}$ object attention value. $z_i \in \{0,1\}^{W \times H}$ is the binary segmentation map[2] of the $i^{th}$ object segmenting the object with the rest of the scene, and has the same dimensions as the image observation. $z_i$ assigns values of 1 for pixels in the image occupied by the $i^{th}$ object, and 0 elsewhere. $T \in [0,1]^{W \times H}$ is the converted state-attention to observation-space attention to act as a target to train the observation-attention network $h_o$ on.

The observation-based module $A_o$ also consists of three networks: $Q_o^\pi$, $\pi_o$, $h_o$ (respectively critic, actor, and attention) with parameters $\{\phi_o, \theta_o, \psi_o\}$. The structure of this module is the same as $A_s$ except the actor and critic now have asymmetric inputs. The input to the actor is the attention-filtered version of the observation, $o_t^a = h_o(o_t) \odot o_t$[3]. The actor and critic networks $\pi_o$ and $Q_o$ are trained with the standard asymmetric DDPG actor and Bellman error losses respectively defined in Section 2.2. The main difference between $A_o$ and $A_s$ is that the observation attention network $h_o$ is trained on both the actor loss and an object-weighted mean squared error loss:

$$\mathcal{L}_{h_o} = E_{o,s \sim \pi_b} \left[ \frac{1}{2} \sum_{ij} \frac{1}{w^{ij}} (h_o(o) - T)^{ij^2} - \nu Q_o(s, \pi_o(o^a)) \right], \quad (4)$$

where weights $w_{ij}$ correspond to the fraction of the partial observation $o$ that the object present in $o^{i,j,1:3}$ occupies, and $\nu$ represents the relative weighting of both loss components. The weight terms, $w$, ensure that the attention network becomes invariant to the size of objects during training and does not simply fit to the most predominant object in the scene. Combining the self-supervised attention loss and the RL loss leverages efficient state-space learning unaffected by visual randomisation.

During training, experiences are collected evenly from both state and observation based agents and stored in a shared replay buffer (similar to Schwab et al. (2019)). This is to ensure that: 1. Both state-based critic $Q_s$ and observation-based critic $Q_o$ observe states that would be visited by either of their respective policies. 2. The attention modules $h_s$ and $h_o$ are trained on the same data distribution to better facilitate alignment. 3. Efficient discovery of highly performing states from $\pi_s$ are used to speed up learning of $\pi_o$.

Algorithm 1 shows pseudocode for a single actor implementation of APRiL. In practice, in order to speed up data collection and gradient computation, we parallelise the agents and environments and ensure data collection from state- and image- based agents is even.

## 4 EXPERIMENTS

To demonstrate the performance and generality of our method, we apply APRiL to a range of environments, and compare with a competitive asymmetric DDPG baseline and various ablations. We evaluate APRiL over different metrics to investigate how attention helps with robustness and generalisation to unseen environments and transfer scenarios. Further experimental details can be found in Appendix C.

### 4.1 EVALUATION PROTOCOL

In order to investigate APRiL under varying conditions, we evaluate in scenarios of increasing complexity covering simple 2D navigation, 3D reaching and 2D dynamic locomotion.

We use the following continuous action-space environments (see Appendix A for further details):

1. *NavWorld*: In this 2D environment, the goal is for the circular agent to reach the triangular target in the presence of distractors. The agent is sparsely rewarded if the target is reached.

2. *JacoReach*: In this 3D environment the goal of the Kinova arm (Campeau-Lecours et al., 2017) agent is to reach the diamond ShapeStacks object (Groth et al., 2018) in the presence of distractors. The agent is rewarded for approaching and reaching its goal.

---

[2]Many simulators, like (Todorov et al., 2012), natively provide functionality to access these segmentations
[3]In practice, the output of $h_o(o_t)$ is tiled to match the number of channels that the image contains

---

**Algorithm 1** Attention Privileged Reinforcement Learning

---

Initialize (a)symmetric actor-critic modules $A_s$, $A_o$, attention alignment module $A_T$, replay buffer $R$
**for** episode= 1 **to** $M$ **do**
   Initial state $s_0$
   **while** $\neg$ DONE **do**
      Render image observation $o_t$ and segmentation maps $z_t$:
            $o_t, z_t \leftarrow \text{renderer}(s_t)$
      **if** episode **mod** $2 = 0$ **then**
         Obtain action $a_t$ using obs-behavioral policy and obs-attention network:
               $a_t \leftarrow \pi_o(h_o(o_t) \odot o_t)$
      **else**
         Obtain action $a_t$ using state-behavioral policy and state-attention network:
               $a_t \leftarrow \pi_s(h_s(s_t) \odot s_t)$
      **end if**
      Execute action $a_t$, receive reward $r_t$, DONE flag, and transition to $s_{t+1}$
      Store $(s_t, o_t, z_t, a_t, r_t, s_{t+1}, o_{t+1})$ in $R$
   **end while**
   **for** $n = 1$ **to** $N$ **do**
      Sample minibatch $\{s, o, z, a, r, s^{'}, o^{'}\}_0^B$ from $R$
      Optimise state- critic, actor, and attention using $\{s, a, r, s^{'}\}_0^B$ with $A_s$
      Convert state-attention to target observation-attention $\{T\}_0^B$ using $\{s, o, z\}_0^B$ with $A_T$
      Optimise observation- critic, actor, and attention using $\{s, o, T, a, r, s^{'}, o^{'}\}_0^B$ with $A_o$
   **end for**
**end for**

---

3. *Walker2D*: In this slightly modified 2D Deepmind Control Suite environment (Tassa et al., 2018) the goal of the agent is to walk forward as far as possible within a time-limit. The agent receives a reward for moving forward as well as a reward for keeping its torso upright.

For these domains we randomise visuals during training as to enable generalisation to these variable aspects of the environment. We randomise a combination of: camera position and orientation, textures, materials, colours, object locations, background. Refer to Appendix B for more details.

## 4.2 KEY RESEARCH QUESTIONS

We investigate the following questions to evaluate how well APRiL accommodates for the transferring of policies across visually distinct environments: Does APRiL 1. Increase **sample-efficiency** during training? 2. Affect **interpolation** performance on unseen environments from the training distribution? 3. Affect **extrapolation** performance on environments outside the training distribution?

We qualitatively analyse the learnt attention maps (both on interpolated and extrapolated domains). Finally, we perform an ablation study to investigate which parts of the APRiL contribute to performance gains. This ablation consists of the following models:

1. *APRiL no self-supervision* (APRiL no sup): APRiL except without the self-supervision provided by the state agent to train the observation-based attention. Both agents are still equipped with an attention module, but the observation attention must now learn without guidance from the state agent. Without bootstrapping from the state agent in this way we expect learning of informative observation-based attention to be hindered.

2. *APRiL no shared buffer* (APRiL no share): APRiL except each agent has its own replay buffer, instead of one shared replay buffer, and hence does not share experiences during training. Under this setting, the observation agent will not be able to benefit from earlier visitation of lucrative states by the state agent. Both agents have an attention module and attention alignment still occurs.

3. *APRiL no background* (APRiL no back): APRiL except the state agent's attention is no longer used to calculate object-space attention values $c$. Instead, all objects are given equal attention and we hence learn a background suppressor. This most competitive ablation investigates how important object suppression is for learning, robustness, and generalisation. Both agents still maintain attention have a shared replay buffer.

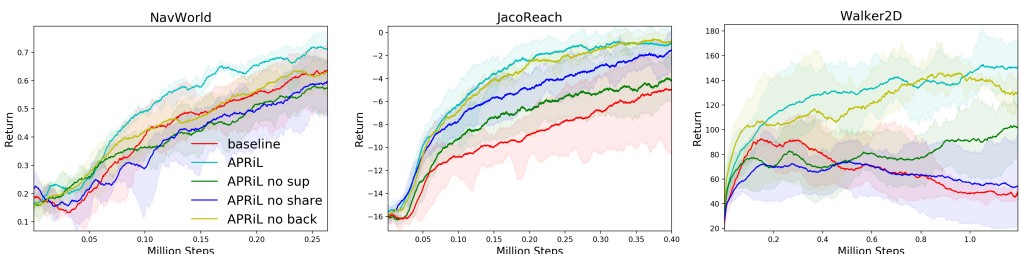

Figure 2: Learning curves during training of APRiL , its ablations, and the asymmetric DDPG baseline. **Solid line**: mean performance. **Shaded region**: covers minimum and maximum performances across 5 seeds.

### 4.3 PERFORMANCE ON THE TRAINING DISTRIBUTION

We evaluate the performance on all domains during training and observe APRiL 's benefits. As seen in Figure 2, APRiL provides performance gains across all continuous action domains. APRiL not only helps learn useful representations quicker (improving learning rate) but also improves final policy performance (within the allotted training time).

The ablations demonstrate that self-supervision and shared replay both independently provide performance gains for *JacoReach* and *Walker2D*[4]. For *Walker2D*, shared replay is crucial as stabilises learning (observe APRiL , APRiL no back, APRiL no sup), due to constant visitation for highly performing states. Suppression of task-irrelevant, yet highly varying, information also speed up learning as simplifies the observation space. For this reason, APRiL no back proves to be a competitive ablation, approaching the performance of APRiL for *JacoReach* and *Walker2D*. For these domains, the background occupies the majority of the observation space and ignoring it already suppresses most of the irrelevant information. Minimal improvement can be achieved by suppressing additional irrelevant objects. None of the ablations, however, are able to outperform the full APRiL framework, demonstrating that the combination of a shared replay buffer and state-space-informed image-attention module cooperate constructively toward more efficient feature learning and effective policy and critic updates.

### 4.4 INTERPOLATION: TRANSFER TO DOMAINS FROM THE TRAINING DISTRIBUTION

We evaluate the performance of all actor-critic algorithms on a hold out set of simulation parameters, unseen during training, from the training distribution. For a detailed description of the training distribution for each domain please refer to Appendix B. For both *NavWorld* and *JacoReach*, the interpolated environments have the same number of distractors, sampled from the same object catalogue, as the training distribution. Table 1 displays final policy performance on these domains. For APRiL , we observe no degradation in policy performance between training and interpolated domains. We see a very similar trend for the asymmetric DDPG baseline. However, as APRiL performs better on the training distribution, its final performance on the interpolated domains is significantly better. We therefore demonstrate that on these domains APRiL's attention mechanism does not hurt with respect to overfitting.

### 4.5 EXTRAPOLATION: TRANSFER TO DOMAINS OUTSIDE THE TRAINING DISTRIBUTION

We investigate performances on simulation parameters outside the training distribution. In particular, we investigate how well APRiL , its ablations, and asymmetric DDPG, generalise to environments with more distractor objects than seen during training. For *NavWorld* and *JacoReach*, we run two sets of increasingly extrapolated experiments with an additional 4 or 8 distractors (refered to as ext-4 and ext-8 in Table 1). The textures and colours of these objects are sampled from a held-old out set of simulation parameters not seen during training. For *NavWorld*, the locations and orientations of the additional distractors are randomly sampled. For *JacoReach*, the locations are

---

[4]We suspect that this is due to the simplicity of *NavWorld*, both visually and due to the small confined state-space, that none of the ablation by themselves outperform the baseline.

sampled from arcs of two concentric circles of different radii (extrapolated arcs and radii to those seen during training), in such a way that each object remains visible. The shapes of the additional distractor object are sampled from the training catalogue of distractor objects. Please refer to Figure 3 for examples of the extrapolated domains.

Table 1 compares performances on the extrapolated sets (except *Walker2D*) varying in difficulty (ext-4 and ext-8). APRiL yields performance gains over the asymmetric DDPG baseline on every extrapolated domain. For *JacoReach*, APRiL's generalisation is so effective that, for the hardest domain with additional 8 distractors, its performance degrades by only 9%[5] opposed to 41% (baseline).

APRiL generalises favorably due to the attention module. Figure 3 shows that attention generalises and suppresses the additional distractors, thereby effectively converting the hold-out observations to those seen during training, which the image-policy can handle. The ablations in Table 1 confirm that in this setting, distractor suppression is crucial. This is seen when comparing the maximum degradation in policy performance of APRiL, APRiL no share, APRiL no back and APRiL no sup (9%, 16%, 27% and 47% respectively). APRiL and APRiL no share both align attention between image and state agents during training, and therefore effectively suppress distractors (yielding a favourable decrease in policy performance of only 9% and 16%). APRiL no back learns a background suppressor, but does not suppress the distractors (leading to a larger degradation of 27%). APRiL no sup has an attention module trained only on the asymmetric actor-critic loss and yields the worst extrapolated performance (47% policy degradation). For these extrapolated domains, the successful suppression of the background **and** additional distractors (achieved only by the full APRiL framework), creates policy invariance with respect to them and helps generalise.

Table 1: Ablation comparing average return over training, interpolated and extrapolated environments (100 each). Results reflect mean and standard deviation of average return over 5 seeds.

| Domain | Baseline | APRiL no sup | APRiL no share | APRiL no back | APRiL |
|---|---|---|---|---|---|
| NavWorld (train) | $0.72 \pm 0.02$ | $0.60 \pm 0.02$ | $0.61 \pm 0.02$ | $0.67 \pm 0.02$ | $\mathbf{0.75 \pm 0.02}$ |
| NavWorld (inter) | $0.70 \pm 0.03$ | $0.60 \pm 0.02$ | $0.62 \pm 0.02$ | $0.69 \pm 0.02$ | $\mathbf{0.80 \pm 0.02}$ |
| NavWorld (ext-4) | $0.45 \pm 0.02$ | $0.42 \pm 0.02$ | $0.43 \pm 0.02$ | $0.45 \pm 0.02$ | $\mathbf{0.53 \pm 0.02}$ |
| NavWorld (ext-8) | $0.46 \pm 0.02$ | $0.42 \pm 0.02$ | $0.38 \pm 0.02$ | $0.47 \pm 0.02$ | $\mathbf{0.50 \pm 0.02}$ |
| JacoReach (train) | $-2.38 \pm 0.23$ | $-3.03 \pm 0.30$ | $-0.10 \pm 0.19$ | $\mathbf{1.10 \pm 0.16}$ | $0.03 \pm 0.19$ |
| JacoReach (inter) | $-2.19 \pm 0.24$ | $-3.38 \pm 0.31$ | $-0.29 \pm 0.20$ | $\mathbf{0.81 \pm 0.18}$ | $0.13 \pm 0.20$ |
| JacoReach (ext-4) | $-6.07 \pm 0.25$ | $-6.81 \pm 0.33$ | $-1.26 \pm 0.22$ | $-0.79 \pm 0.23$ | $\mathbf{-0.70 \pm 0.23}$ |
| JacoReach (ext-8) | $-8.02 \pm 0.24$ | $-9.26 \pm 0.28$ | $-2.69 \pm 0.24$ | $-3.53 \pm 0.26$ | $\mathbf{-1.47 \pm 0.25}$ |
| Walker2D (train) | $67.4 \pm 3.15$ | $107 \pm 3.31$ | $63.0 \pm 2.13$ | $126 \pm 4.33$ | $\mathbf{155 \pm 3.25}$ |
| Walker2D (inter) | $68.9 \pm 3.25$ | $107 \pm 3.30$ | $62.6 \pm 2.11$ | $126 \pm 4.31$ | $\mathbf{156 \pm 3.20}$ |

## 4.6 ATTENTION MODULE ANALYSIS

To better comprehend the role of the attention, we visualise APRiL's attention maps (Figure 3, 4, 5) on both interpolated and extrapolated domains. For *NavWorld*, attention is correctly paid to all relevant aspects (agent and target; circle and triangle respectively). Attention generalises reasonably well to the extrapolated environments. For *JacoReach*, attention looks at the target, diamond-shaped, object as well as every other link (alternating links) of the Kinova arm. Interestingly, APRiL learnt that as the arm is a constrained system, the state of every other link can be indirectly inferred without explicit attention. The state of the unobserved link can be inferred by observing the links either side of it. The entropy loss over the state-attention module encourages this form of attention over minimal set of objects. Attention here generalises very well to the extrapolated domains. For *Walker2D*, we observe attention that is dynamic in object space. The attention module attends different subsets of links depending on the state of the system (see Figure 5). When the walker is upright, walking, and collapsing, APRiL pays attention to the lower limbs, every other link, and foot and upper body, respectively. We suspect that in these scenarios, the magnitude of the optimal action depends on the state of and as is largest for the lower links (due to stability), every link (coordination), and foot and upper body (large torque required), respectively.

---

[5]Percentage decrease is taken with respect to initial and final policy performance on training distribution

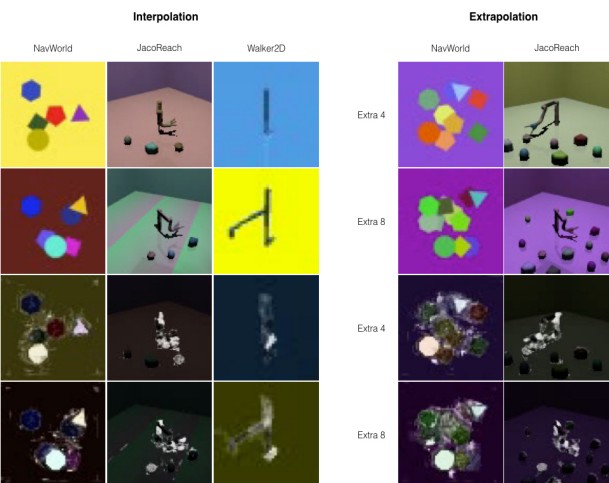

Figure 3: Example held-out domains (top) and APRiL attention maps (bottom). White and black signify high and low attention values. Attention correctly suppresses background and distractors.

## 5 RELATED WORK

**Domain Randomisation** has been applied for reinforcement learning to facilitate transfer between domains (Tobin et al., 2017; Pinto et al., 2018; Sadeghi & Levine, 2016; Viereck et al., 2017; OpenAI et al., 2018; Held et al., 2017) and increase robustness of the learned policies (Rajeswaran et al., 2016). However, while domain randomisation enables us to generate more robust and transferable policies, it leads to a significant increase in required training time (OpenAI et al., 2018).

Existing comparisons in the literature demonstrate that, even without domain randomisation, the increased dimensionality and potential partial observability complicates learning for RL agents (Tassa et al., 2018; Schwab et al., 2019; Watter et al., 2015; Lesort et al., 2018). In this context, accelerated training has been achieved by using **access to privileged information** such as environment states to asymmetrically train the critic in actor-critic RL (Schwab et al., 2019; Pinto et al., 2018). In addition to using additional information to train the critic, Schwab et al. (2019) use a **shared replay buffer** for data generated by image- and state-based actors to further accelerate training for the image-based agent. Our method extends these approaches by sharing information about relevant objects by aligning agent-integrated attention mechanisms between an image- and state-based actors.

Recent experiments have demonstrated the strong dependency and bidirectional interaction between attention and learning in human subjects (Leong et al., 2017). In the context of machine learning, **attention mechanisms** have been integrated into RL agents to increase robustness and enable interpretability of an agent's behaviour (Sorokin et al., 2015; Choi et al., 2017; Mott et al., 2019). In comparison to these works, we focus on utilising the attention mechanism as an interface to transfer information between two agents to enable faster training.

## 6 CONCLUSION

We introduce Attention Privileged Reinforcement Learning (APRiL), an extension to asymmetric actor-critic algorithms that leverages access to privileged information like exact simulator states. The method benefits in two ways, via sharing a replay buffer as well as aligning attention masks between image- and state-space agents. By leveraging simulator ground-truth information about system states, we are able to learn efficiently in the image domain especially during domain randomisation where feature learning becomes increasingly difficult. Our evaluation on a diverse set of environments demonstrates significant improvements over the competitive asymmetric DDPG baseline and reveals that APRiL learns to generalise favourably to environments not seen during training (both within and outside of the training distribution) in comparison to the strong baseline; emphasising the importance of attention and shared experience for robustness of the learnt policies.

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

# Appendix

## A    ENVIRONMENTS

1. *NavWorld*: In this sparse reward, 2D environment, the goal is for the circular agent to reach the triangular target in the presence of distractor objects. Distractor objects have 4 or more sides and apart from changing the visual appearance of the environment cannot affect the agent. The state space consists of the $[x, y]$ locations of all objects. The observation space comprises RGB images of dimension $(60 \times 60 \times 3)$. The action space corresponds to the velocity of the agent. The agent only obtains a sparse reward of $+1$ if the particle is within $\epsilon$ of the target, after which the episode is terminated prematurely. The maximum episodic length is 20 steps, and all object locations are randomised between episodes.

2. *JacoReach*: In this 3D environment the goal of the agent is to move the Kinova arm (Campeau-Lecours et al., 2017) such that the distance between its hand and the diamond ShapeStacks object (Groth et al., 2018) is minimised. The state space consists of the quaternion position and velocity of each joint as well as the Cartesian positions of each ShapeStacks object. The observation space comprises RGB images and is of dimension $(100 \times 100 \times 3)$. The action space consists of the desired relative quaternion positions of each joint (excluding the digits) with respect to their current positions. Mujoco uses a PD controller to execute 20 steps that minimises the error between each joint's actual and target positions. The agent's reward is the negative squared Euclidean distance between the Kinova hand and diamond object plus an additional discrete reward of $+5$ if it is within $\epsilon$ of the target. The episode is terminated early if the target is reached. All objects are out of reach of the arm and equally far from its base. Between episodes the locations of the objects are randomised along an arc of fixed radius with respect to the base of the Kinova arm. The maximum episodic length is 20 agent steps.

3. *Walker2D*: In this 2D modified Deepmind Control Suite environment (Tassa et al., 2018) with a continuous action-space the goal of the agent is to walk forward as far as possible within 300 steps. We introduce a limit to episodic length as we found that in practice this helped stabilise learning across all tested algorithms. The observation space comprises of 2 stacked RGB images and is of dimension $(40 \times 40 \times 6)$. Images are stacked so that velocity of the walker can be inferred. The state space consists of quaternion position and velocities of all joints. The absolute positions of the walker along the x-axis is omitted such that the walker learns to become invariant to this. The action space is setup in the same way as for the *JacoReach* environment. The reward is the same as defined in (Tassa et al., 2018) and consists of two multiplicative terms: one encouraging moving forward beyond a given speed, the other encouraging the torso of the walker to remain as upright as possible. The episode is terminated early if the walker's torso falls beyond either $[-1, 1]$ radians with the vertex or $[0.8, 2.0]$m along the z axis.

## B    RANDOMISATION PROCEDURE

In this section we outline the randomisation procedure taken for each environment during training.

1. *NavWorld*: Randomisation occurs at the start of every episode. We randomise the location, orientation and colour of every object as well as the colour of the background. We therefore hope that our agent can become invariant to these aspects of the environment.

2. *JacoReach*: Randomisation occurs at the start of every episode. We randomise the textures and materials of every ShapeStacks object, Kinova arm and background. We randomise the locations of each object along an arc of fixed radius with respect to the base of the Kinova arm. Materials vary in reflectance, specularity, shininess and repeated textures. Textures vary between the following: noisy (where RGB noise of a given colour is superimposed on top of another base colour), gradient (where the colour varies linearly between two predefined colours), uniform (only one colour). Camera location and orientation are also randomised. The camera is randomised along a spherical sector of a sphere of varying radius whilst always facing the Kinova arm. We hope that our agent can become invariant to these randomised aspects of the environment.

3. *Walker2D*: Randomisation occurs at the start of every episode as well as after every 50 agent steps. We introduce additional randomisation between episodes due to their increased duration. Due to the MDP setup, intra-episodic randomisation is not an issue. Materials, textures, camera location and orientation, are randomised in the same procedure as for *JacoReach*. The camera is setup to always face the upper torso of the walker.

## C    IMPLEMENTATION DETAILS

Table 2: Model architecture. FC() represents a (multi-layered) fully connected network with the number of nodes per layer stated as argument. Conv() represents a (multi-layered) convolutional network whose arguments take the form [channels, square kernel size, stride] for each hidden layer.

| Domain | NavWorld and JacoReach | Walker2D |
|---|---|---|
| State Actor | FC($[256]$) | FC($[256]$) |
| Obs Actor | Conv($[[18, 7, 1], [32, 5, 1], [32, 3, 1]]$) | Conv($[[18, 8, 2], [32, 5, 1], [16, 3, 1], [4, 3, 1]]$) |
| State Critic | FC($[64, 64]$) | FC($[400, 300]$) |
| Obs Critic | FC($[64, 64]$) | FC($[400, 300]$) |
| State Attention | FC($[256]$) | FC($[256]$) |
| Obs Attention | Conv($[[32, 8, 1], [32, 5, 1], [64, 3, 1]]$) | Conv($[[32, 8, 1], [32, 5, 1], [64, 3, 1]]$) |
| Replay Buffer Size | $10^4$ | $2 \times 10^5$ |

In this section we provide more details on our training setup. Refer to table 2 for the model architecture for each component of APRiL and the asymmetric DDPG baseline. *Obs Actor* and *Obs Critic* setup are the same for both APRiL and the baseline. *Obs Actor* model structure comprises of the convolutional layers (without padding) defined in table 2 followed by one fully connected layer with 256 hidden units (FC($[256]$)). All layers use ReLU (Romero et al., 2015) activations and layer normalisation (Ba et al., 2016) unless otherwise stated. Each actor network is followed by a tanh activation and rescaled to match the limits of the environment's action space.

The *State Attention* module includes the fully connected layer defined in table 2 followed by a Softmax operation. The *Obs Attention* module has the convolutional layers (with padding to ensure constant dimensionality) outlined in table 2 followed by a fully connected convolutional layer (Conv($[1, 1, 1]$)) with a Sigmoid activation to ensure the outputs vary between $0$ and $1$. The output of this module is tiled in order to match the dimensionality of the observation space.

During each iteration of APRiL (for both $A_o$ and $A_s$) we perform 50 optimization steps on mini-batches of size $64$ from the shared replay buffer. The target actor and critic networks are updated every iteration with a Polyak averaging of $0.999$. We use Adam (Kingma & Ba, 2014) optimization with a learning rate of $10^{-3}$, $10^{-4}$ and $10^{-4}$ for critic, actor and attention networks respectively. We use default TensorFlow (Abadi et al., 2016) values for the other hyperparameters. The discount factor, entropy weighting and self-supervised learning hyperparameters are $\gamma = 0.99$, $\beta = 0.0008$ and $\nu = 1$ respectively. To stabilize learning, all input states are normalized by running averages of the means and standard deviations of encountered states.

Both actors employ adaptive parameter noise (Plappert et al., 2017) exploration strategy with initial std of $0.1$, desired action std of $0.1$ and adoption coefficient of $1.01$. The settings for the baseline are kept the same as for APRiL where appropriate.

# D   ATTENTION VISUALISATION

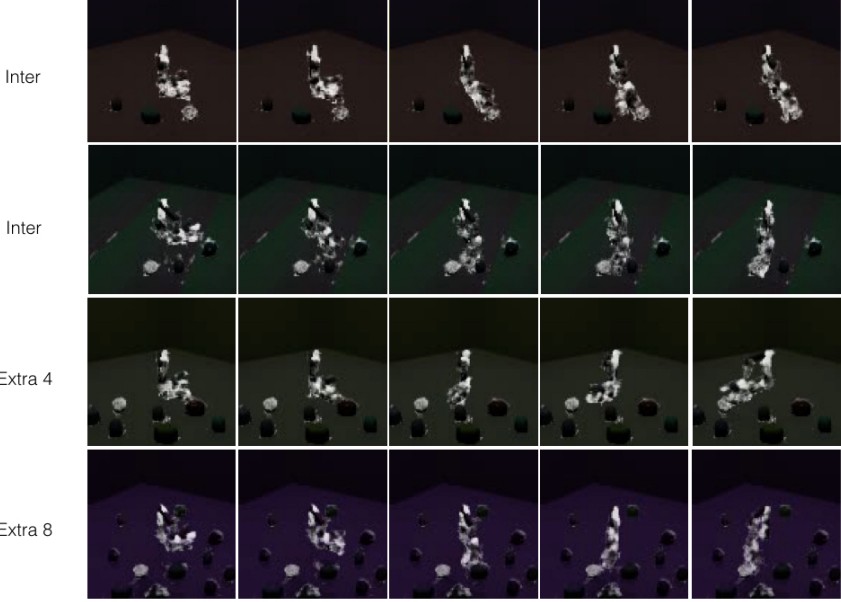

Figure 4: APRiL attention maps for policy rollouts on NavWorld and Jaco domains. White and black signify high and low attention values respectively. For NavWorld and JacoReach, attention is correctly paid only to the relevant objects (and Jaco links), even for the extrapolated domains. Refer to section 4.6 for more details.

**Walker2D**

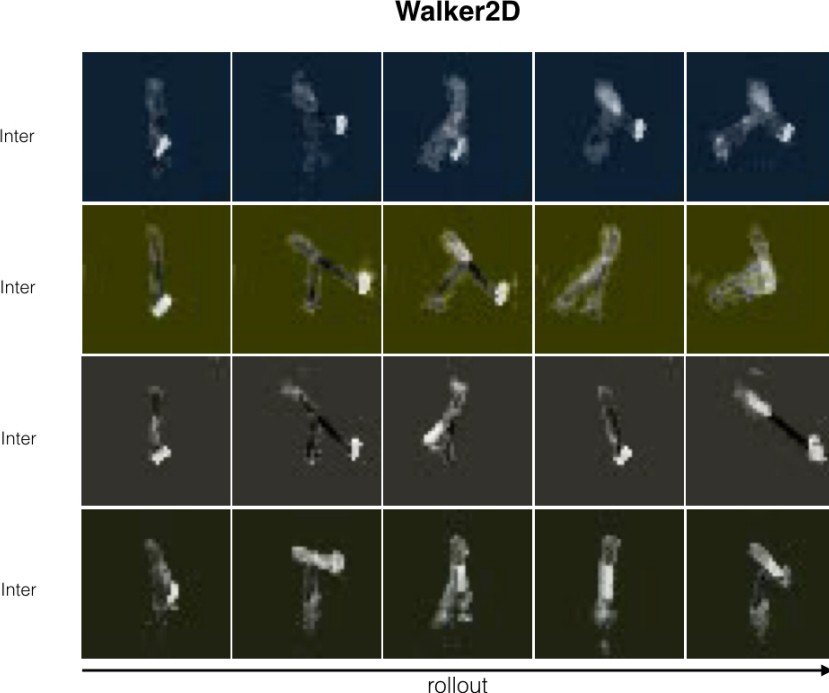

Figure 5: APRiL attention maps for policy rollouts on Walker domain. White and black signify high and low attention values respectively. Attention varies based on the state of the walker. When the walker is upright, high attention is paid to lower limbs. When walking, even attention is paid to every other limb. When about to collapse, high attention is paid to the foot and upper torso. Refer to section 4.6 for more details.

