# OpenReview forum: "Attention Privileged Reinforcement Learning for Domain Transfer"
_ICLR.cc/2020/Conference — Reject_

### Official Review · AnonReviewer3 · 2019-10-21
**Official Blind Review #3**

**Rating:** 1

**Review:**

The topic addressed by the paper is domain adaptation and transfer learning in the text context of deep reinforcement learning, in particular the “sim2real” problem, where a policy is learned in simulation and should be transferred to a physical agent in a real-world scenario. The work builds on the existing “asymmetric DDPG” formulation (Pinto et al., 2017), which exploits the fact that full states are sometimes available in simulated environments but not during deployment. In Pinto et al., this is addressed by learning an actor taking as input observations, and a critic which has access to the state.

The contribution of the paper is an extension of Pinto et al. by adding additional communication between the state head (which learns a critic) and the observation head (which learns the policy). This is done through an attention mechanism, now very classical in deep learning, which weights the different elements of the (state) input. The particularity here is that the attention mechanism is trained on the critic, which takes the full state as input, and then transferred to the observed input of the policy. The problem is that the state is very different from the observations, which are images. An alignment module expands the distribution over state variables to a distribution over observed objects.

One of my main concerns here is lack of justification and lack of clarity. While the abstract of the paper and introduction section are well written, and wants us to learn more about this interesting idea, the paper kind of falls apart in the subsequent chapters. The authors mention alignment of attention, but the exact motivation for the algorithm, i.e. the motivation for its formulation, are never provided. The description itself is also far from clear, as key design choices are not introduced and we discover them, or guess them, from equations. We don’t know, for instance, what the observations are, and think that they are images. However, apparently an object mask detector is run over the input images, as segmentation maps are input to the state-object alignment module, which needs to distribute the attention over states to attention over objects.

This also means that the formulation is quite specific to the task at hand, since it seems to be object-centric.

Figure 1 is another example of the same problem: There are many arrows, but their signification is unclear. Dashed lines are indicated to mean alignment, but alignment is not a standard term, for instance comparable to a computation connection, or a loss signal.

A similar confusion is found in the equations themselves, as h(s) is mapped to a vector c, but this vector decomposes into different objects of the environment. What is T? It being upper case could mean a matrix or a scalar value, but we are not sure. From the loss function we see that this seems to be vector (a distribution) of the same size as the attention vector h_o … which is over what?

Basically, as I understand it, the alignment loss minimizes some structured mapping between some attention vector over objects on the observation side and some attention vector over states on the critics side. This seems to be a weak loss signal, as the attention transformation needs to be learned together with the attention mechanism itself (and of course the actor and the critic).

At the beginning of section 3, the method is introduced of having an asymmetric part and a symmetric part, but I don’t see this, as this would require learning 4 predictors and not 2 (an actor and a critic). An asymmetric model, as given in Pinto et al., uses two predictors, an actor and a critic, with different loss functions and, more importantly, different inputs. Here, the authors claim that the observation module itself is asymmetric … but how can something by asymmetric if it contains only an actor? Same, the critic is supposed to be symmetric … but without an actor?

As I see it, w.r.t to the question whether the model is symmetric or asymmetric, the formulation is identical to Pinto et al., thus asymmetric. The difference lies in the attention mechanism and the alignment module.

At some point in the paper, this method is mentioned to be self-supervised … I am not sure which aspect of this work could be called self-supervised, but I agree that the definition of this relatively new term is sometimes ambiguous.

On the other hand, I think the method should be compared to “classical” self-supervision in RL, which corresponds to predicting information which is available during training but not available as input (depth prediction etc.). Here, a natural baseline seems to be a symmetric model (same input for actor and critic: observations) and to predict the state of the model and self-supervise it during training but not deployment.

The evaluation is unfortunately not convincing.

Two important baselines are missing:
-	Self-supervision, as mentioned above
-	Pinto et al., on which this method is based.

There are some hics in the results … for instance the curves in Figure 2 show learning in progress, they are not yet converged. Also, the standard deviations are quite big.

Another downside is that, although sim2real is used as a motivation for this paper (since this is the most typical scenario where states are available during training but not during testing), the experiments have not been performed using physical agents. Testing is performed on degraded simulated agents. I do understand that physical environments are harder to manage than simulated ones, but tiny physical environments can be obtained for reasonable prices, and this would have made the paper much stronger and more aligned with its core motivation. Simulated noise cannot replace a physical environment.

No details have been given on the additional distractor objects unseen during training.

The Walker 2D environment is described as “modified”, but how exactly, and why?

Section 2.1 (definition of POMDPs) seems to be unnecessary and can be deleted.


**Experience Assessment:**

I have published one or two papers in this area.

**Review Assessment: Checking Correctness Of Derivations And Theory:**

I carefully checked the derivations and theory.

**Review Assessment: Checking Correctness Of Experiments:**

I carefully checked the experiments.

**Review Assessment: Thoroughness In Paper Reading:**

I read the paper thoroughly.

---

> ### Author Response · Authors · 2019-11-14
> **Response to Reviewer 3**
>
> We thank the reviewer for the detailed and insightful comments, which have greatly improved the paper. We’ll be glad to answer any further questions.
>
> The main feedback is around the motivation, justification, written clarity; the relation to asymmetric actor-critic; and the inclusion of baselines.
> We have significantly improved the clarity of writing to address the former; but there appears to be a miscommunication around the latter two: our approach employs an actor and critic for both state and observation - based agents, and Asymmetric Actor-critic (suggested as a baseline by the reviewer) is the main baseline against which we compare. We have repeated this in the text to ensure this is clear to all readers, and thank the reviewer for highlighting this confusion.
>
> We address the reviewer’s detailed points systematically below.
>
> - The contribution of the paper is an extension of Pinto et al. by adding additional communication between the state head (which learns a critic) and the observation head (which learns the policy)
> …
> At the beginning of section 3, the method is introduced of having an asymmetric part and a symmetric part, but I don’t see this, as this would require learning 4 predictors and not 2 (an actor and a critic).
>
> There seems to be a misunderstanding here: both the observation and state agents have an actor and a critic (so four predictors in total). This is shown in Figure 1, and mentioned in the introduction and in Section 3, where we state that the state-based module uses symmetric DDPG, while the observation-based module uses asymmetric DDPG.
>
> - At some point in the paper, this method is mentioned to be self-supervised … I am not sure which aspect of this work could be called self-supervised, but I agree that the definition of this relatively new term is sometimes ambiguous.
>
> We use the term self-supervision to refer to the use of the state agent’s attention map as target for the attention of the observation agent. This is stated in Section 3, and we have updated the other sections of paper to repeat this definition for clarity.
>
>
> - Two important baselines are missing:
>   -	Self-supervision, as mentioned above
>   -	Pinto et al., on which this method is based.
>
> Pinto et al. is in fact the main baseline we compare against in the paper; we modified section 4 and changed the description in the plots to make this clear. The proposed self-supervised baseline (learned a mapping to state) is interesting, but cannot  generalise to held-out environments with additional distractor objects without explicitly specifying the objects that are relevant to the task. In contrast, APRiL does not have this requirement, as attention is used to prune out objects that are not relevant to the task.
>
> - No details have been given on the additional distractor objects unseen during training.
>
> Thank you for pointing out this concern. We have modified section 4.5 to explicitly describe in more detail the experimental setup of the extrapolated domains. In summary, the extrapolation domains correspond to environments with additional 4 or 8 more distractor objects than seen during training. The colour, locations, textures of these objects are obtained from a held-out set not seen before. The shapes are sampled from the training-distribution object catalogue. Please refer to the following link for a visualisation of these domains https://sites.google.com/view/april-domain-randomisation/home.
>
> - The Walker 2D environment is described as “modified”, but how exactly, and why?
>
> Appendix A describes how walker2d was modified and why. The modification was made to the maximum number of episodic steps. We introduced a lower limit as we found that in practice this helped stabilize learning across all tested algorithms including the baseline.
>
> - The attention transformation needs to be learnt
>
> The attention transformation performed by the attention alignment module AT is in-fact predefined by the adjacency matrix M and object segmentation maps z. It therefore does not have to be learnt. We have modified section 3 to clarify this for the readers.

---

> > ### Comment · AnonReviewer3 · 2019-11-14
> > **After rebuttal**
> >
> > I thank the authors for the rebuttal, which did address some of my questions on issues like "symmetric vs. asymmetric", and the Pinto baseline.
> >
> > However, several strong objections have not been addressed, among which are
> > - experiments (sim2real without "real")
> > - Unfinished training (no convergence yet)
> > - writing and presentation

---

> > > ### Author Response · Authors · 2019-11-15
> > > **Remaining Concerns of Reviewer 3**
> > >
> > > We are glad to have addressed some of the reviewer’s questions.
> > >
> > > We address the reviewer’s remaining concerns below.
> > >
> > > - experiments (sim2real without "real")
> > >
> > > Although we have not been able to run experiments on a physical robot, we have demonstrated APRiL’s benefits for learning rate during domain randomisation across 3 diverse domains. The domain randomisation approach we perform is the same as that in Pinto et al. so there is reason to believe our approach would be beneficial for sim-to-real transfer. We have also demonstrated favorable generalisation to extrapolated domains with additional distractors. We believe we provide a valid intellectual contribution to the field of robust, generalisable autonomous agents.
> > >
> > > - Unfinished training (no convergence yet)
> > >
> > > We regret that the reviewer is not satisfied with the learning curves convergences. We still demonstrate, however, that APRiL considerably speeds up learning and generalises significantly better to extrapolated domains (9% as opposed to 41% performance degradation) due to its self-supervised attention module. We believe both these contributions still hold and are beneficial for the field.
> > >
> > > - writing and presentation
> > >
> > > We have made multiple changes to significantly improve the written clarity of the paper and address the previous misunderstandings flagged by the reviewer. Would the reviewer be able to let us know what remaining specific concerns they have regarding the writing and presentation?

---

> > > > ### Comment · AnonReviewer3 · 2019-11-15
> > > > **After rebuttal**
> > > >
> > > > You cannot evaluate generalization if the training curves did not converge.
> > > >
> > > > I gave detailed explanations on writing and presentation in my review, it does not look like they have been addressed (as also most of my questions had been ignored in the first rebuttal).

---

> > > > > ### Author Response · Authors · 2019-11-15
> > > > > **Response to Reviewer 3**
> > > > >
> > > > > We thank the reviewer again for their prompt response. We have worked hard to address all of the posed questions and are honestly unable to identify what questions have not been answered. We made various changes to the method section to clarify the terminology which the reviewer found unclear. Regarding design choices, we motivate our approach in the introduction and method sections. We also clearly define attention alignment in the method section which should help for interpreting Figure 1.
> > > > >
> > > > > We believe we have systematically accounted for all of the questions raised by the reviewer, but if they could elaborate on which points they feel have been unaddressed (feel free to simply copy individual previous questions) we will be happy to point to where we have addressed them, or work on fixing them.

---

### Official Review · AnonReviewer2 · 2019-10-23
**Official Blind Review #2**

**Rating:** 3

**Review:**

Overall the method is interesting but I am not overly convinced the method provides a significant improvement over simpler methods that are not compared to the work here. Also, the generalization and extrapolation experiments need more description. As well, the work talks about how this method can be used to accelerate learning for transfer to real-robotic systems but does not have an example of this.

More detailed comments:
- When you are referencing a paper that is very similar to your method and you build off of it is good to link to the published version ( the reference for Asymmetric DDPG was published at RSS2018.)
- While the method is very interesting I feel that one of the obvious baselines that would be good to try is to train a model that maps o -> s (image to state). This is a very simple method and appears to accomplish the goals for the authors. A discussion on why this would not be good enough and some results to show that this performs poor would be good addition.
- Related to the last point why are object-specific segmentation maps needed? It seems like the method alone was not capable enough to learn from pixels alone. Also, Can these segmentation maps be visually described? It is not very clear. Where do the segmentation maps come from?
- Can you describe the shared replay buffer in more detail? What does no shared replay buffer mean?
- The caption in Figure 2 is rather sparse and comes before the explication of the different methods in the figure. This figure needs more explanation. What are the other versions of April? What is the baseline? There does not appear to be a significant improvement? Can you show more of a qualitative improvement via videos of the policy performance?
- It appears the method does not work as well on the Atari games can the authors provide some insights into why the method does not offer as much of an improvement? Is there no Attentive DQN for Pong, natural?
- I do not understand what experiment is being performed in section 4.3.2. Maybe it would help if you explained what a canonical image is? Where do you get the compressed state information?
- In 4.4 you show that APRiL does not do better than the baseline? What is the baseline?
- In section 4.5 you describe how the method generalizes to the task with additional distractors. Can you describe specifically how this experiment is designed? How do the distractors compare to the other objects in the scene? are their colors changed? How many are there? This section needs a lot more detail to understand how much the reader can surmise about the methods ability to generalize.
- In the same section, the authors say the image segmentation is crucial. Does this imply that maybe we should just put an image segmentation network in between the observation and the policy inputs? What is using this additional attention mechanism better than that simpler method?
- For Table 1: It is interesting that the baseline does better than the APRiL no share or no sup. It might be important to perform an exploration as to why having those features reduces performance and the baseline appears more robust.
- It would be nice to see this work applied to more tasks. For now the reacher and walking are the most interesting but those tasks are also somewhat basic. How would this method work on the walker 3d where there is even more partial observation?
- Can the authors explain more how they have deduced from the attention map that the attention has figured out how to indirectly discover where the Jaco links are without explicit attention? I do not see this.
- For figure 4, how the attention is visualized could be explained better. Are the white regions receiving all the attention? Also, What makes the difference between a Interpolation example and a Extrapolation example? Is it just more objects in the scene?
- The authors should also reference "Sim-to-Real Transfer of Robotic Control with Dynamics Randomization".

**Experience Assessment:**

I have published in this field for several years.

**Review Assessment: Checking Correctness Of Derivations And Theory:**

I carefully checked the derivations and theory.

**Review Assessment: Checking Correctness Of Experiments:**

I carefully checked the experiments.

**Review Assessment: Thoroughness In Paper Reading:**

I read the paper at least twice and used my best judgement in assessing the paper.

---

> ### Author Response · Authors · 2019-11-14
> **Response to Reviewer 2**
>
> Thank you for the thoughtful and constructive feedback. Addressing your comments has considerably improved the submission. Please do reach out if new questions arise.
>
> The main feedback is around the written clarity regarding; segmentation maps, shared replay buffer, baselines, ablations, held-out domains, attention maps. We have significantly improved the clarity of writing in the paper to address all of these topics.
>
> We address the reviewer’s points systematically below.
>
> - Why not perform image-state mapping?
>
> This is interesting, but cannot generalise to held-out environments with additional distractor objects without explicitly specifying the objects that are relevant to the task. In contrast, APRiL does not have this requirement, as attention is used to prune out objects that are not relevant to the task.
>
> - What and how are the object segmentation maps obtained?
>
> An individual object segmentation map is a binary segmentation of the image, segmenting that object with the rest of the scene. Mujoco simulator provides functionality to obtain these segmentations. For more details please refer to Section 3 which has been modified to further clarify object segmentation maps.
>
> - Could you explain further the shared replay buffer?
>
> Similar to Schwab et al. 2019, we use one shared replay buffer between agents. Each agent appends its own experiences to the same buffer. During training, both actor critic modules sample experiences uniformly from the shared replay. We have added this to Section 3; please refer to the pseudocode for further clarity.
>
> - Figure 2, provide more informative captions. What are/is the ablations of APRiL, baseline, shared no buffer? Which ablations have attention?
>
> The baseline refers to asymmetric DDPG that we introduce in section 2.2. We have made this clearer in the paper. All experiment architectures are covered in Appendix C. We made improvements to the description of the ablations in section 4.2. Each ablation has state and image attention. However, only APRiL and APRiL no shared buffer perform attention alignment. For APRiL no shared buffer each agent has its own replay buffer instead of sharing.
>
> - What didn’t work for APRiL Atari? What is the state?
>
> For the original APRiL Atari setup, the state corresponded to a non-randomised image (no background randomisation). In the lead up to the rebuttal, we identified additional research questions for this setup. Therefore APRiL Atari was omitted to improve the paper’s clarity by focussing on the control domains. This remains an interesting future direction which we will investigate further. Given multiple reviewers requested algorithm and experimental setup clarity, we used the remaining space to address these points.
>
> - Section 4.4, APRiL does not do better than baseline.
>
> Table 1 shows that the full APRiL architecture (final column) does perform better than the asymmetric DDPG baseline (second column) for all interpolation experiments ((inter) in the domain column).
>
> - Better explain the extrapolation setup. What is the difference between this and interpolation domain?
>
> We have added a detailed description of the extrapolated and interpolated domains in sections 4.4 and 4.5. In summary, the extrapolated domains correspond to environments with additional distractors. Attributes are obtained from an unseen held-out set. Shapes are sampled from the training-distribution. The interpolation domain has no additional distractors. Domain visualisation is seen here https://sites.google.com/view/april-domain-randomisation/home.
>
> - If image segmentation crucial why not add an image segmentation network between observation and policy inputs?
>
> We believe you are referring to section 4.5: ‘distractor suppression is crucial’ for generalisation. Distractor suppression requires more than just segmenting. It requires learning which segments and objects correspond and which are irrelevant and should be suppressed. Only the entire APRiL framework achieves this. We have clarified this in Section 4.5.
>
> - Tasks are simple. How would this method work for walker 3d?
>
> Due to domain randomisation the tasks are significantly trickier to solve than usual. The setup for walker3d would be the same walker2d. Attention should be similar to the 2d counterpart. For this harder domain, we suspect that the shared replay will be very beneficial due to the larger explorable state-space efficiently visited by the state-agent.
>
> - Explain how attention is paid to every other link and how can other states be inferred? Explain attention maps in Figure captions.
>
> We have updated captions to clarify the attention maps. White and black signify high and low attention values. Attention is paid to alternating links. The state of the link not attended can be inferred by attending links either side of it. We have added a detailed description to section 4.5. The following videos that may aid with interperability https://sites.google.com/view/april-domain-randomisation/home.

---

> > ### Comment · AnonReviewer2 · 2019-11-14
> > **Thank you for your clarifications**
> >
> > Thank you for your clarifications
> >
> > These comments and the paper updates have helped clear up my understanding of some important details.
> >
> > You mention that the website has information on the attention map and visualize it. It is not clear on the website which videos are the attention maps. If the labelling could be improved it would help understand what each of the videos is on the webpage.

---

> > > ### Author Response · Authors · 2019-11-14
> > > **Website Update**
> > >
> > > We thank the reviewer for their quick response and advice regarding how the website can be improved. We have added descriptions to the webpage to describe the policy rollouts for both APRiL and the Asymmetric DDPG baseline, as well as the attention maps.
> > >
> > > We hope this helps clarify the videos for the reviewer.

---

### Official Review · AnonReviewer1 · 2019-10-24
**Official Blind Review #1**

**Rating:** 3

**Review:**

Summary - Building on top of the domain randomization principle (used to train policies robust to domain-variations) to learn policies which transfer well to new domains, the paper proposes an approach to improve and speed-up learning / training over randomized environments. The paper operates in a settings where the policy to be transferred only has access to observations -- images, etc -- and not the complete underlying state of a (simulated) environment. The underlying idea is to -- (1) maintain two sets of actor-critic networks - a symmetric pair where the actor has access to the underlying state and an asymmetric pair where the actor has access only to image observations; (2) evenly gather experiences from behavioral policies of both actors and store them in a shared replay buffer and (3) learn to align the attention placed by the policies over objects in the environment for the state and observation based actors. The idea is to leverage privileged information about the state (which is strictly more informative compared to observations) to learn robust observation based policies. Experimental results indicate the proposed approach improves generalization performance compared to several ablations of the same on both in-distribution and out-of-distribution environments.

Strengths

- The paper is generally well-written and easy to follow. The authors generally do a good job of motivating the proposed approach by leveraging access to privileged information in the environment / simulation / renderer during training to get more robust observation policies to transfer to novel settings. The proposed approach is presented after appropriately grounding the problem setting and preliminaries and the authors clearly state and evaluate on the axes of research questions they care about.

- The proposed approach is somewhat novel and extends prior work on using shared replay buffers and asymmetric actor critic methods to accelerate training. Although the specific focus is on aligning object-level attention from a state-based actor and an observation based actor, the authors adopt design choices that help in preventing degenerate solutions -- for instance, the object-weighted squared error loss component to learn the observation attention module.

- The experimental results more-or-less support the claims of the paper (however, only in comparison with ablations and 1 baseline) in the sense we see improvements in terms of average returns and sample-efficiency plots. Furthermore, the authors conduct ablations to understand which components of the proposed approach contribute significantly in different environments -- for both extrapolated and interpolated environments.

- Sec 4.6 presents interesting analysis of the learned attention (observation) attention mechanism for both interpolated and extrapolated environments. It seems that attention is generally placed over relevant aspects (objects / links) of the environment and the associated takeaways seem feasible.

Weaknesses

- Having said that, there some weaknesses / questions which if addressed would make the paper stronger and help in increasing the rating of the paper.

- Access to the object-specific attention maps seems to be an assumption that might not scale well across simulators. More realistic / richer simulations (say, 3D reconstructions of indoor rooms) may not always offer this much privileged information -- one might have access to fine-grained reconstructions from multiple viewpoints, but not object level maps. This, combined with the fact that major gains have only been demonstrated over the specified continuous control domains (ignoring the Atari results), makes me slightly concerned about the scalability of the proposed approach in terms of more real-world + applicable domain-transfer scenarios. Maybe a more general approach that learns to match some intermediate representations of the state and observation based actors is a more general approach. Can the authors comment on this?

- Including the entropy loss while training the attention mechanism for the state based actor is justified only through feasible interpretations of the attention visualizations for the JacoReach experiments. I’m curious how important is it for the state-based attention to be sparse? Does it actually affect performance if the entropy loss is not included while learning the state-based attention module?

- Although the paper mentions experience is gathered evenly over the behavioral policies of both the actors it’s slightly unclear how that is being performed without actually referring to the algorithm pseudocode in the appendix. I would encourage the authors to include / move the same to the main paper. It makes the entire pipeline much easier to grasp.

- Transfer to novel environments (not necessarily with domain-shifts) has also been studied in context of providing exploration incentives (see InfoBot - https://arxiv.org/abs/1901.10902) in addition to a (sparse / dense) episodic reward. I would be curious to see how well does APRiL compare to such approaches in a setting where both are applicable - say the MultiRoomNXSY set of experiments in InfoBot (pointed above). This is to understand if the gains obtained from APRiL are very specific to domain-shifts or are largely applicable to any form of novel environment transfer. Can the authors comment on this?

Reasons for rating

Beyond the above points of discussion, I don’t have major weaknesses to point 	out. I generally like the paper. The authors do a good job of identifying the sliver in which they make their contribution and motivate the same appropriately. My major point of concern is centered is around the fact that APRiL (probably) assumes access to much privileged information which may not be generally available across all kinds of environments, etc. My rating of the paper is based on the strengths and weaknesses highlighted above. Addressing / Responding to those appropriately would definitely help in improving the rating of the paper.

**Experience Assessment:**

I have read many papers in this area.

**Review Assessment: Checking Correctness Of Derivations And Theory:**

I assessed the sensibility of the derivations and theory.

**Review Assessment: Checking Correctness Of Experiments:**

I assessed the sensibility of the experiments.

**Review Assessment: Thoroughness In Paper Reading:**

I read the paper thoroughly.

---

> ### Author Response · Authors · 2019-11-14
> **Response to Reviewer 1**
>
> We thank the reviewer for the thoughtful and constructive feedback. It has greatly helped improve the paper. We hope the following responses address all concerns.
>
> The main feedback is around justification of model setup, scalability of the approach, and written clarity. We have moved the pseudocode to section 3 and rewritten certain sections of the paper to address these concerns.
>
> We address the reviewer’s detailed points systematically below.
>
> - Assumed access to segmentation maps is restricting. How about a more scalable approach that learns to match some intermediate representation between image and state actors?
>
> It is true that not all simulators will enable easy access to object level segmentation maps. However, with enough access to the rendering system, provided by many simulators such as Mujoco, it is not hard to obtain them. We thank the reviewer for the interesting suggestion for a more scalable approach. APRiL-DQN for Atari, which for reasons mentioned in the ‘rebuttal summary’ section has been removed from the paper and left for future research, attempts to match attention representations between ‘state’ and ‘image’ agents without the need for segmentation maps. We see the proposed approach by the reviewer as an extension to this idea. If segmentation maps are available, however, this as simply another source of privileged information which we exploit in order to speed up learning of the image agent.
>
> - How important is the entropy loss for the state-based attention? Is it important for the attention to be sparse?
>
> Preliminary experiments on NavWorld and JacoReach showed that an entropy loss on the state-attention module was helpful in producing sparse object-space attention. This lead to better learning rate and generalisation for the agent (due to superior distractor suppression). We now mention this in the paper in section 3.
>
> - Including the pseudocode in the main paper will clarify how to evenly gather experiences.
>
> Thank you for this useful suggestion. We have moved the pseudocode to section 3 to clarify this for the readers.
>
> - How well would APRiL compare to InfoBot? Are APRiL’s benefits specific to domain-shifts or are they largely applicable to any form of novel environment transfer?
>
> APRiL’s benefits are clear for domain randomisation, where the input to the observation agent’s policy is highly varying with a lot of redundant information, necessitating self-supervised attention and generally simulations which provide additional low-dimensional information. However, extending APRiL to exploration in new domains is an interesting avenue for future research and we thank the reviewer for this suggestion.

---

> > ### Comment · AnonReviewer1 · 2019-11-15
> > **Thanks for responding to the comments!**
> >
> > Thanks to the authors for responding to my comments in detail and making the appropriate changes in the draft. I'll respond to the author's comments below.
> >
> >
> > >It is true that not all simulators will enable easy access to object level segmentation maps. However, with enough access to the rendering system, provided by many simulators such as Mujoco, it is not hard to obtain them. We thank the reviewer for the interesting suggestion for a more scalable approach. APRiL-DQN for Atari, which for reasons mentioned in the ‘rebuttal summary’ section has been removed from the paper and left for future research, attempts to match attention representations between ‘state’ and ‘image’ agents without the need for segmentation maps. We see the proposed approach by the reviewer as an extension to this idea. If segmentation maps are available, however, this as simply another source of privileged information which we exploit in order to speed up learning of the image agent.
> >
> > My concern in the comment was not centered around what is possible in the simulators being used in the paper but more around the broader applicability of the proposed approach beyond such simulators. In my opinion, assuming access to segmentation maps generally seems equivalent to assuming access to an unreasonable amount of privileged information.
> >
> > Thanks for reporting the observations obtained regarding entropy loss and updating the paper to reflect the same.
> >
> > >APRiL’s benefits are clear for domain randomisation, where the input to the observation agent’s policy is highly varying with a lot of redundant information, necessitating self-supervised attention and generally simulations which provide additional low-dimensional information. However, extending APRiL to exploration in new domains is an interesting avenue for future research and we thank the reviewer for this suggestion.
> >
> > My concern in the comment was not necessarily regarding extending APRiL to exploration in new domains, but more along the lines of trying to disentangle transfer performance in presence / absence of a domain shift. Thanks for responding to this though.

---

### Author Response · Authors · 2019-11-14
**Rebuttal Summary and Atari Update**

We thank the reviewers for their constructive feedback, which has helped to significantly strengthen the paper. The main overall changes include:

- A much clearer explanation and justification of the baselines (asymmetric actor-critic and ablations).
- Further explanation of the difference some of the key architectural details and some of the design decisions made (such as on object segmentation maps, attention alignment, etc).
- A number of additional changes to aid clarity of the paper.

We address each of the reviewers’ specific concerns in the direct comments and hope our improvements have made the paper clearer for future readers. In addition, we encourage reviewers to review the changes directly in the updated submission and revision history.

Finally, we identified relevant additional research questions related to adapting APRiL to the Atari domain. In this context, we decided to omit Atari from the paper and improve the paper’s overall clarity by focussing on the 3 continuous control domains which originally inspired the development of APRiL. This remains an interesting future direction (how can we exploit APRiL for robustness in Atari and other domains without lower-dimensional state information?) which we will investigate further. Given the request from multiple reviewers for additional information about the algorithm and experimental setup, we have used the remaining space to address these points.

---

### Decision · Program_Chairs · 2019-12-19

**Decision:**

Reject

**Comment:**

This paper tackles the problem of transferring an RL policy learned in simulation to the real world (sim2real). More specifically, the authors address the situation where the agent can access privileged information available during simulation, for example access to exact states instead of compressed representations. They perform experiments in various simulated domains where different aspects of the environment are modified to evaluate generalization.

Major concerns remain following the rebuttal. First, it is not clear how realistic it is to assume access to such privileged information in practice. Second, the experiments are not convincing since the algorithms do not appear to have reached convergence in the presented results. Finally, a sim2real work would highly benefit from real-world experiments.

In light of the above issues, I recommend to reject this paper.